# Temporal Flexibility in Spiking Neural Networks: A Novel Training Method for Enhanced Generalization Across Time Steps

## Abstract

Spiking Neural Networks (SNNs), models inspired by neural mechanisms in the brain, allow for an energy-efficient implementation on neuromorphic hardware. However, the limitation of current direct training approaches lies in their ability to only optimize parameters for an SNN operating at a specific time step. This leads to the necessity for fine-tuning when generalizing to additional time steps, resulting in considerable computational inefficiency. In this study, we initially examine the feasibility of parameter sharing across structurally identical SNNs operating at different time steps. Subsequently, we propose an innovative training methodology-mixed time step training (MTT) that facilitates the development of a temporal flexible SNN (TFSNN). Throughout the training process, various time steps are arbitrarily assigned to distinct SNN blocks, accompanied by the establishment of novel inter-block communication protocols. Following training, the TFSNN can be simplified to an SNN operating at any chosen fixed time step, eliminating the need for fine-tuning. Experimental results across all primary datasets demonstrate that the TFSNN exhibits robust generalization capabilities surpassing existing training methodologies reliant on a fixed time step. Notably, we achieved a 96.84% accuracy rate on the CIFAR10 dataset, an 81.98% accuracy rate on the CIFAR100 dataset, and a 68.34% accuracy rate on the ImageNet dataset with T = 6.

## 1 Introduction

The field of deep learning continues to evolve and has witnessed groundbreaking advancements such as GPT-4 (Bubeck et al., 2023) and SAM (Kirillov et al., 2023) that have made unprecedented strides across diverse applications. However, the deployment of these large-scale neural networks on low-power edge devices presents substantial challenges. In addition to the typical solutions including network quantization (Rastegari et al., 2016), trimming (He et al., 2017), and distillation (Hinton et al., 2015), the Spiking Neural Network, known as one of the 3rd generation of neural networks, has emerged as a compelling candidate due to its unique bio-inspired characteristics (Fang et al., 2021; Guo et al., 2022; Yao et al., 2023). SNNs mimic the behavior of biological neurons by accumulating membrane potentials and transmitting sparse pulses, thereby circumventing the need for computationally expensive multiplications (Roy et al., 2019). This unique property of SNNs opens up new avenues for efficient computation on neuromorphic hardware (Davies et al., 2018; Khan et al., 2008; Akopyan et al., 2015), presenting a promising solution for energy-efficient neural computation.

Nevertheless, despite their advantages, training SNNs presents distinct challenges due to the binary nature of their spiking behavior. This prevents the direct application of traditional gradient descent methods. Current practices, such as the surrogate gradient (SG) method (Wu et al., 2018), aim to address this issue by replacing the gradient of a non-differentiable step function with that of an approximate function, thus enabling a back-propagation process similar to RNNs (Zheng et al., 2021; Deng et al., 2022; Xiao et al., 2022; Chen et al., 2023; Meng et al., 2023). However, these methods are predominantly designed to find optimal parameters for the SNN at a specific time step, necessitating computationally expensive fine-tuning when the learned models are transferred to those with distinct temporal latency.

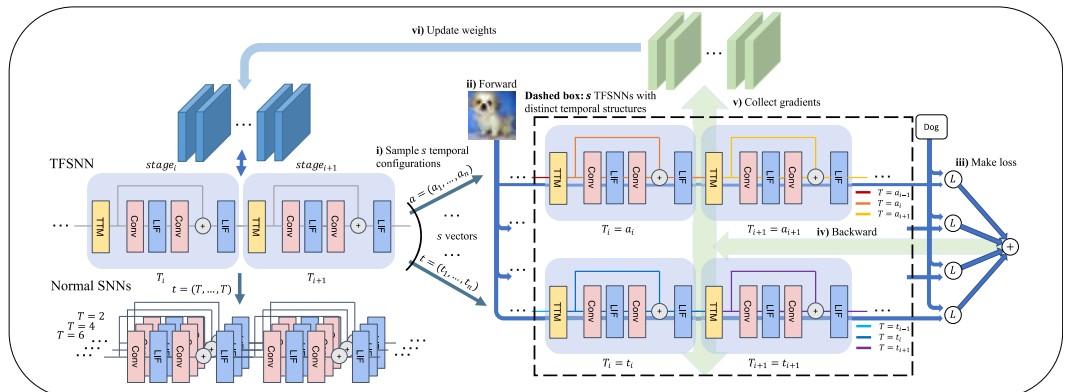

**Figure 1:** The workflow of MTT pipeline. In each iteration, we sample $s$ temporal configurations, each assigning random time steps to different stages. These configurations create $s$ TFSNNs with distinct temporal structures, all sharing the same weights. Next, we forward propagate the $s$ TFSNNs and compute their respective losses. To update the shared weights, we backpropagate the sum of the $s$ losses to obtain the gradient. After training, the TFSNN can transform into a normal SNN and exhibit high performance at any time step.

In response to this issue, we propose a novel training method-mixed time step training (MTT) to acquire the temporal flexible SNN (TFSNN). The workflow is shown in Fig. 1. This method secures a set of parameters adaptive to SNNs with varying time steps under the same network structure, thereby extending the generalization capabilities of SNNs across different time steps and circumventing the need for fine-tuning across different time steps. Our main contributions are as follows:

- We first demonstrate the feasibility of training an SNN that is compatible with different time steps using a simple training method, naive mixture training (NMT).

- We analyze the mechanism by which NMT improves the performance of the model and propose to train a TFSNN by mixed time step training (MTT) based on this mechanism to further improve the training method. We also propose a method to estimate the accuracy of TFSNN for any time step combination.

- We conducted thorough testing on both static and event-based datasets and on a wide range of architectures, to confirm the effectiveness of our approach. The neural network obtained by MTT demonstrates impressive flexibility over time, achieves performance on par with other state-of-the-art methods, and exhibits more friendliness with asynchronous chips.

## 2 RELATED WORK

**ANN-SNN Conversion** ANN-SNN conversion uses SNN firing rates to approximate the activation of ANN. Specifically, parameters are first directly copied from a pre-trained ANN to the target SNN and then fine-tuned to mimic the original ANN activation. Techniques have been proposed to reduce the minimum convertible time step, including the subtraction mechanism (Rueckauer et al., 2016; Han et al., 2020; Han & Roy, 2020), spike-norm (Sengupta et al., 2019), threshold shift (Deng & Gu, 2021), layer-wise calibration (Li et al., 2021a) and activation quantization (Bu et al., 2023). Although SNNs obtained by conversion show some temporal flexibility for large time steps (e.g., above 100), they do not exhibit temporal flexibility under ultra-low time steps. Also, the conversion method cannot handle DVS datasets and can only procure SNNs with IF neurons.

**Direct Training** The direct training approach stems from the idea that SNNs can be viewed as variant RNNs and trained using BPTT as long as the non-differentiable activation term is replaced with a surrogate gradient. Wu et al. (2018) first propose the STBP method and train SNNs using an ANN framework. Zheng et al. (2021) further suggest a novel BatchNorm strategy tdBN to facilitate large-scale SNN training. Recently, the performance of SNN on neuromorphic datasets has been substantially enhanced with the advent of specially developed algorithms including TET (Deng et al., 2022) and TCJA-SNN (Zhu et al., 2022). On static datasets, various methods (Li et al., 2021b; Guo et al., 2022; Yao et al., 2022) are proposed to close the gap between SNNs and ANNs. Notably, Guo et al. (2022) first report SNNs with accuracy exceeding the corresponding ANN counterpart,

demonstrating the strong potential of SNNs. However, weights obtained by existing direct training methods are only applicable to a specific time step, which entails further training when the inference time step is different.

**Dynamic Inference Time Step** Recent research explores inference-wise varied time steps, to reduce average inference cost by skipping steps when the network is confident enough. Li et al. (2023b) introduced SEENN, which determines the exit time step using confidence scores (SEENN-I) or a policy network (SEENN-II). Li et al. (2023a) introduced another confidence-based dynamic model, identifying the optimal confidence threshold using a Pareto Front. The excellent temporal flexibility of TFSNN satisfies the need for such methods to reason at less than the training time step and thus is an ideal model provider for them. We demonstrate this by experiments in section 5.2.

## 3 PRELIMINARIES

### 3.1 SPIKING NEURON MODEL

We adopt the iterative Leaky and Integrate-and-Fire(LIF) model (Wu et al., 2018; 2019). The membrane potential is updated as

$$\boldsymbol{v}(t+1) = \tau \boldsymbol{u}(t) + \boldsymbol{I}(t), \tag{1}$$

$$\boldsymbol{I}(t) = \mathbf{W} \cdot \boldsymbol{x}(t), \tag{2}$$

where $\boldsymbol{u}(t)$ denotes the membrane potential of the time step $t$, $\tau$ is a constant leaky factor, and $\boldsymbol{I}(t)$ is the pre-synaptic inputs given by the product of synaptic weight $\mathbf{W}$ and spiking input $\boldsymbol{x}(t)$. After the membrane potential exceeds a threshold $V_{th}$, the neuron fires a spike and resets $\boldsymbol{u}$ to 0. The hard reset mechanism and firing function can be expressed as

$$\boldsymbol{s}(t+1) = \boldsymbol{\Theta}(\boldsymbol{v}(t+1) - V_{th}), \tag{3}$$

$$\boldsymbol{u}(t+1) = \boldsymbol{v}(t+1) \cdot (1 - \boldsymbol{s}(t+1)), \tag{4}$$

where $\boldsymbol{\Theta}(\cdot)$ denotes the Heaviside step function, $\boldsymbol{s}(t+1)$ is the spike that will propagate to the next layer. In this work, all the experiments are conducted with $V_{th}$ set to 1 and $\tau$ set to 0.5.

### 3.2 SURROGATE GRADIENT

The direct training method computes gradients for parameters by spatiotemporal backpropagation (Wu et al., 2018):

$$\frac{\partial L}{\partial \mathbf{W}} = \sum_t \frac{\partial L}{\partial \boldsymbol{s}(t)} \frac{\partial \boldsymbol{s}(t)}{\partial \boldsymbol{v}(t)} \frac{\partial \boldsymbol{v}(t)}{\partial \boldsymbol{I}(t)} \frac{\partial \boldsymbol{I}(t)}{\partial \mathbf{W}}. \tag{5}$$

When backpropagating, all terms apart from the term $\frac{\partial \boldsymbol{s}(t)}{\partial \boldsymbol{v}(t)}$ can be easily calculated. However, the term $\frac{\partial \boldsymbol{s}(t)}{\partial \boldsymbol{v}(t)} = \frac{\partial \boldsymbol{\Theta}(v)}{\partial v}$ is the derivative of the Dirac delta function and does not exist. To solve this problem, surrogate gradient(SG) is used to approximate the original gradient. In this work, we adopt a triangular surrogate gradient (Rathi & Roy, 2020), which can be formulated as

$$\frac{\partial \boldsymbol{s}(t)}{\partial \boldsymbol{v}(t)} = \frac{1}{h^2} \max(0, h - |V_{th} - \boldsymbol{v}(t)|), \tag{6}$$

where h is a constant controlling the sharpness. In this work, we apply h=1 to all experiments.

## 4 METHODOLOGY

In this section, we introduce our method Mixed Timestep Training (MTT), and its derivation process. We first experimented with a simple method Naive Mixture Training (NMT) in Section 4.1 to verify the possibility of bringing SNNs temporal flexibility. NMT not only succeeded in enabling SNNs to generalize across different time steps but also consistently outperformed every single SDT-trained model at each time step. In Section 4.2, we delve into the reasons behind NMT's performance improvement, examining both gradient and generalization factors. Subsequently, in Section 4.3, we present our final method, MTT—an enhanced version of NMT that produces Temporal Flexible SNN (TFSNN) with high adaptiveness to varied time steps.

## 4.1 START FROM NAIVE MIXTURE TRAINING

Our approach commences with a fundamental concept of altering the SNN's simulation time step during the training process. To avoid confusion, we refer to the above training strategy as "naive mixture training" (NMT) and the typical training strategy as "standard direct training" (SDT). **Notes that NMT is not our final method**. In the NMT technique, we randomly sample $s$ values from the pre-selected time set (e.g. $\{1, 2, 3, 4, 5, 6\}$) for each mini-batch. Each sampled $T_i$ value is assigned as the simulation time step of SNN for the current forwarding process. The network's parameters are updated only after all the $s$ forwarding processes are finished. An experiment on the CIFAR100 dataset was conducted to analyze the effect of NMT. For SDT, we train the SNN for five independent runs with different time steps, whereas for NMT, we only train the SNN once and test it under five different time step configurations. We also assess a single T=6 SNN accuracy trained with SDT across different T values for comparison. The results are presented in Table 1.

Our approach with the proposed simple strategy NMT shows improved performance across all time step settings. This outcome underscores the practicality of training an SNN that is compatible with various time steps and motivates us to delve deeper into understanding the associated mechanisms.

**Table 1:** Inference accuracy of naive mixture training vs. standard direct training. SDT* denotes independently trained SNNs with SDT

| Methods | T=2 | T=3 | T=4 | T=5 | T=6 |
|---------|-------|-------|-------|-------|-------|
| SDT | 70.08 | 72.77 | 74.17 | 75.09 | 75.63 |
| SDT* | 72.86 | 73.86 | 74.77 | 74.96 | 75.63 |
| NMT | 73.47 | 74.17 | 75.11 | 75.34 | 75.77 |

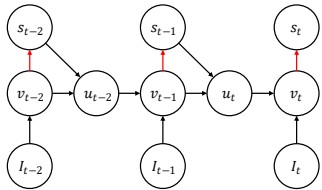

**Figure 2:** Computation graph of LIF neuron, red arrow denotes non-differentiable step function $\Theta$.

## 4.2 ANALYSIS ON NAIVE MIXTURE TRAINING

**Gradient.** We believe that there exists a balance between SNN's output precision and gradient accuracy during the training phase with a surrogate gradient. Here, "output precision" refers to the information encoding capability of the spike trains. Due to the binary nature of the spike-based inter-layer information transmission in SNNs, their information encoding capacity is not as strong as that of ANNs. For example, a spike train composed of three spike values (each value can be 0 or 1) can only express $2^3$ possible values. while the information encoding capacity of high-precision numerical data is almost unlimited ($2^{32}$). Fortunately, spike trains (SNN layer output) have both spatial and temporal dimensions, and expanding these dimensions can enhance the total information encoding capacity, i.e., "output precision". The spatial dimension is constrained by the network structure, while the temporal dimension is controlled by the SNN time step T. Increasing the time step T can lead to an improvement in the information encoding capacity, thereby enhancing the "output precision" of SNN. "Gradient accuracy" refers to the precision of the parameter gradients obtained by backpropagation. The reason for this notion arises from the fact that in the direct training of SNNs, the gradients computed by the backpropagation algorithm contain noise. We first derive the backpropagation formula Eq. 7 for further analysis. Based on equations Eq. 1-4, the computational graph of a neuron with T time steps can be represented as shown in Fig. 2. This representation enables us to compute the derivative of each output $s$ to each input $I$ (For further elaboration, refer to appendix A.5):

$$\frac{\partial s(t)}{\partial I(t-n)} = \frac{\partial s(t)}{\partial v(t)} \cdot \tau^n \prod_{i=t-n}^{t} \left[(1 - s(i)) - v(i) \cdot \frac{\partial s(i)}{\partial v(i)}\right] \tag{7}$$

where $n$ is a non-negative integer that denotes the time span between the specified $s$ and $I$. Note that $\partial s(t)/\partial I(p)$ with $p > t$ is constantly zero because of the invalid propagation path. The backpropagation noises of SNN originate from the term $\frac{\partial s(i)}{\partial v(i)}$ in Eq. 7, where this term is calculated as an approximated value by surrogate gradient during backpropagation because its original function is a step function, whose derivative is ill-defined. As shown in Eq. 7, when the time steps T increases,

there are more instances of $\frac{\partial s(i)}{\partial v(i)}$, leading to more gradient noise in the calculated parameter gradient, which sets an extra barrier to SNN training. To sum up, increasing the time step (T) enhances the SNN's output precision, but it also results in an increase in the calculated surrogate gradient noise. This partly explains why SNN's performance cannot be infinitely improved by increasing time steps. Similarly, although decreasing the time step sacrifices output precision, the gradient is more accurate.

A related work (Meng et al., 2023) suggests effectively training an SNN by removing the temporal direction gradients. Similar to this work, we discovered that the gradients obtained under different time steps (T) settings are remarkably similar, with over 94% cosine similarity between the gradients acquired from scenarios with $T = 3$ and $T = 6$. NMT benefits from the accurate gradient of low T while maintaining output precision of high T and therefore does the best of both worlds. We verify our theory of the accurate gradient brought by low T through experiments in section 5.3.

**Generalization**. If SNN training falls into a local minimum point, once NMT samples a new time step that is far away from the current one, the SNN's output may change significantly. In this scenario, the new training loss may not converge, leading SNN to jump out of the local minimum point. And eventually, the SNN will be trained towards a flatter minimum point. Another perspective is that since the sampling space is 6, NMT is equivalent to training 6 similar SNNs simultaneously. This is similar to applying a new kind of dropout to the SNN, which improves the network's generalization. This is why NMT significantly improves SNN's performance when T is small. We verify our theory through experiments in section 5.3 and loss landscapes in appendix A.10.

### 4.3 TEMPORAL FLEXIBLE SPIKING NEURAL NETWORK

Inspired by NMT, we develop a novel SNN called temporal flexible Spiking Neural Network (TF-SNN): A normal SNN is partitioned into stages as shown in Fig. 3 (e.g., a ResNet block as a stage), and each stage can sample different simulation time steps. So, using the same time step select set, TFSNN expands the network sample space of NMT from 6 to $6^8$ on the ResNet-18 architecture. And the NMT sample space is a special subset of TFSNN. Then, we design the temporal transformation module (TTM) to modify the communication rules between pre- and post-blocks with different simulation time steps. Finally, we develop a new training pipeline, mixed time step training (MTT), to train the TFSNN. It is worth noting that a trained TFSNN can unify the simulation time steps of different stages and degrade into a traditional SNN.

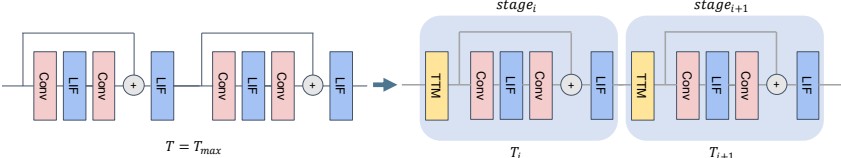

**Figure 3:** Evolve a normal SNN into a TFSNN.

In TFSNN, the setting of time steps of these stages can be denoted by a temporal configuration vector $\boldsymbol{t} = (t_1, \ldots, t_n)$, where $t_i$ denotes the time step of the $i$-th stage and $n$ is the total number of stages. The forwarding of a TFSNN can be denoted by $S_{TF}(\boldsymbol{x}, \boldsymbol{t})$, where $\boldsymbol{x}$ is an input and $\boldsymbol{t}$ is the temporal configuration vector.

### 4.3.1 TEMPORAL TRANSFORMATION MODULE

We first design the basis of TFSNN, an inter-block communication rule between blocks with different simulation time steps, based on which we develop the temporal transformation module (TTM). TTMs can be categorized into downsampling and upsampling types as illustrated in Fig. 4. In downsampling TTMs, we borrow from the pooling layer and divide in-

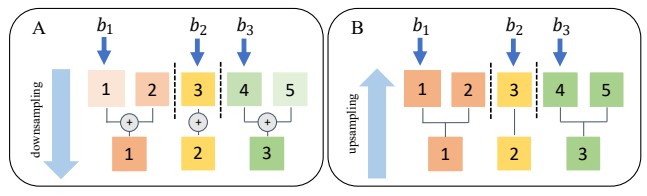

**Figure 4:** (A) Downsampling TTM when $t_{in}$=5 and $t_{out}$=3. (B) Upsampling TTM when $t_{in}$=3 and $t_{out}$=5.

put time frames into $t_{out}$ groups of adjacent frames. We then sum up the frames within each group to form $t_{out}$ time frames. By contrast, the upsampling type of TTM replicates each input time frame and assigns it to all output frames in the corresponding group, using the grouping policy same as the downsampling type of TTM with $t_{out}$ frames of input and $t_{in}$ frames of output.

The task at hand is to identify a suitable policy to partition $l$ frames into $k$ groups, where $l \geq k$. A natural idea is to group them evenly. So, we use the following policy shown in Eq. 8 where $b_i$ denotes the index of the first frame of group $i$. We further explain our design in appendix A.4.

$$b_i = \lfloor \frac{(i-1) \cdot l}{k} - \varepsilon \rceil + 1, i \in \{1, ..., k\} \qquad (8)$$

### 4.3.2 MIXED TIME-STEP TRAINING

**Overall MTT Framework** We further develop mixed time step training (MTT) to train a TFSNN. Mathematically, our goal is to minimize the overall loss:

$$\mathcal{L}_{MTT(overall)} = \sum_k^N \sum_{\boldsymbol{t} \in \{T_{min}, ..., T_{max}\}^l} \mathcal{L}(S_{TF}(\boldsymbol{x}_k, \boldsymbol{t}), \boldsymbol{y}_k) \qquad (9)$$

where $\mathcal{L}$ is any loss function, $N$ is batch size, $l$ is the number of stages, $T_{min}$ and $T_{max}$ are the minimum and maximum time steps respectively, $S_{TF}$ is a TFSNN, and $\boldsymbol{t}$ is any possible temporal configuration vector. Since directly optimizing the overall loss is too expensive, we sample $s$ vectors $\boldsymbol{t}_1, \ldots, \boldsymbol{t}_s \in \{T_{min}, \ldots, T_{max}\}^l$ for each iteration and optimize the estimated loss function instead:

$$\mathcal{L}_{MTT} = \sum_k^N \sum_{\boldsymbol{t} \in \{\boldsymbol{t}_1, ..., \boldsymbol{t}_s\}} \mathcal{L}(S_{TF}(\boldsymbol{x}_k, \boldsymbol{t}), \boldsymbol{y}_k) \qquad (10)$$

To better illustrate our method, the training pipeline of one epoch is detailed in Algo.1. In this work, $T_{min}$ is set to 1 for all experiments.

---

**Algorithm 1** Mixed time step training for one epoch

---

**Input:** SNN model $S_{TF}$; training dataset; training iteration $I$; sample number $S$ in one iteration; block number $N$; minimum and maximum time step $T_{min}, T_{max}$
 1: **for** all $i = 1, 2, \ldots, I$-th iteration **do**
 2:   Get the training data $\boldsymbol{x}_i$ and labels $\boldsymbol{y}_i$
 3:   **for** all $s = 1, 2, \ldots, S$-th sample **do**
 4:    Sample a vector $\boldsymbol{t}_s$ with $N$ numbers in the range $[T_{min}, T_{max}]$
 5:    Calculate the loss function $\mathcal{L}(S_{TF}(\boldsymbol{x}_i, \boldsymbol{t}_s), \boldsymbol{y}_i)$
 6:    Backpropagation and collect the gradient
 7:   **end for**
 8:   Update the model weights with collected gradients
 9: **end for**

---

**Batch Normalization Calibration** MTT implemented with the standard BN technique suffers significant accuracy degradation. This is because when training with mixed time steps, drastic structural changes lead to significant variations in batch statistics. Therefore, the running mean and running variance calculated during training would be inaccurate for a network trained with MTT. To address this problem, we estimate BN statistics from a few training batches after other parameters are well-trained and fixed. In our experiments in section 5.4, correcting BN statistics with as few as 10 batches proved sufficient. Therefore, we use this setting in all experiments. We also discovered that the BN statistics of $T_{max}$ apply to other time steps. This enables us to avoid extra calibration when switching to a different inference $T$. See details in appendix A.7.

### 4.3.3 ACCURACY ESTIMATION

Although our proposed method is initially intended to develop an SNN adaptable to all unified inference time steps, obtaining the accuracy of any temporal configuration $\boldsymbol{t}$ can help us identify high-performing, low-energy TFSNN settings and even inspires SNN structure designs. However, it is

impossible to test all possible situations directly (e.g., there are $6^8$ cases for ResNet-18). Therefore, we propose the following hypotheses to estimate the accuracy of each $t$ : 1) The expressive power of each block in TFSNN contributes differently and positively to the final network accuracy. 2) The expressive power of each block is related to the information content of its selected time T, such as $K\sqrt{\log_2 T}$, where the square root is due to the information content of a spike sequence with time step T cannot exceed $\log_2 T$ because the arrangement of spikes is regular, e.g., tending to be uniformly distributed. Then we assume that the accuracy equation of TFSNN is $\sum_{i=1}^{I} K_i \sqrt{\log_2 T_i} + c$, where the $K_i$ is the contribution of each block, and c is a constant. Our experiment (see section 5.3) demonstrates that we only need to sample a very small number of $t$s to infer the parameters of the equation, and then able to estimate all TFSNN accuracy with this equation.

## 5 EXPERIMENTS

In this section, we first compare our method with other current training methods to demonstrate the effectiveness of our method in training temporal flexible SNNs. Then, we conduct some validation and ablation experiments to prove that our method improves the network's generalization and the effectiveness of the different parts of our method. The datasets involved in this work include static datasets like CIFAR10, CIFAR100 (Krizhevsky et al., 2009), and ImageNet (Deng et al., 2009), and event-based datasets such as CIFAR10-DVS (Li et al., 2017) and N-Caltech101 (Orchard et al., 2015). We also tested our method on sequence task and audio task (see appendix A.12). The model structures used in this paper include ResNet-18 (He et al., 2016), ResNet-19 (Zheng et al., 2021), ResNet-34 (He et al., 2016), VGG series (see appendix A.1 for VGG experiments), and our handmade structures (see appendix A.6). All the experiments were conducted with RTX3090 GPUs.

**Table 2:** Compare with existing works on static image datasets. † denotes introducing additional floating-point multiplications

| Dataset | Model | Methods | Architecture | TimeStep | Accuracy |
|---|---|---|---|---|---|
| CIFAR10 | Guo et al.(Guo et al., 2022) | InfLoR-SNN | ResNet-19 | 6 | 96.49±0.08 |
| | | | | 4 | 96.27±0.07 |
| | | | | 2 | 94.44±0.08 |
| | Deng et al.(Deng et al., 2022) | TET | ResNet-19 | 6 | 94.50±0.07 |
| | | | | 4 | 94.44±0.08 |
| | | | | 2 | 94.16±0.03 |
| | Yao et al.(Yao et al., 2022) | GLIF† | ResNet-19 | 6 | 95.03±0.08 |
| | | | | 4 | 94.85±0.07 |
| | | | | 2 | 94.44±0.10 |
| | **Our Method** | MTT | ResNet-19 | 6 | **96.84**±0.03 |
| | | | | 4 | **96.75**±0.04 |
| | | | | 2 | **96.20**±0.07 |
| CIFAR100 | Li et al.(Li et al., 2021b) | Dspike | ResNet-18 | 6 | 74.24±0.10 |
| | | | | 4 | 73.35±0.14 |
| | Guo et al.(Guo et al., 2022) | InfLoR-SNN | ResNet-19 | 6 | 79.51±0.11 |
| | | | | 4 | 78.42±0.09 |
| | Deng et al.(Deng et al., 2022) | TET | ResNet-19 | 6 | 74.72±0.28 |
| | | | | 4 | 74.47±0.15 |
| | Yao et al.(Yao et al., 2022) | GLIF† | ResNet-19 | 6 | 77.35±0.07 |
| | | | | 4 | 77.05±0.14 |
| | **Our Method** | MTT | ResNet-19 | 6 | **81.98**±0.03 |
| | | | | 4 | **81.51**±0.04 |
| ImageNet | Zheng et al. (Zheng et al., 2021) | STBP-tdBN | ResNet-34 | 6 | 63.72 |
| | Deng et al. (Deng et al., 2022) | TET | ResNet-34 | 4 | 64.79 |
| | Fang et al. (Fang et al., 2021) | SEW† | SEW-ResNet-34 | 4 | 67.04 |
| | Chen et al. (Chen et al., 2023) | MPSNN† | DSNN-34 | 4 | 67.52 |
| | | FSNN | FSNN-34 | 4 | 66.45 |
| | **Our Method** | MTT | ResNet-34 | 6 | **68.34** |
| | | | | 4 | **67.54** |

### 5.1 COMPARISON TO EXISTING WORKS

Here, we compare our TFSNN trained by MTT with existing works. To demonstrate the superior temporal flexibility of TFSNN, for one backbone and one dataset in the table, we trained only once. For all experiments, we apply the sampling number $s = 3$ unless otherwise specified. The results of static and neuromorphic datasets are provided in Table 2 and Table 3. To enable a fair comparison with existing works, our TFSNN will degrade into a standard SNN with a unified time step after training, so the inference process of TFSNN will be exactly the same as existing works. We repeat

**Table 3:** Compare with existing works on DVS datasets. †denotes introducing additional floating-point multiplications

| Dataset | Model | Methods | Architecture | T | Accuracy |
|---|---|---|---|---|---|
| CIFAR10-DVS | Yao et al. (Yao et al., 2022) | GLIF† | 7B-wideNet | 16 | 78.10 |
| | Guo et al. (Guo et al., 2022) | InfLoR-SNN | ResNet-19 | 10 | 75.50±0.12 |
| | Zhu et al. (Zhu et al., 2022) | TCJA-SNN† | VGGSNN | 10 | 80.7 |
| | Deng et al. (Deng et al., 2022) | TET | VGGSNN | 10 | **83.17**±0.15 |
| | **Our Method** | MTT | ResNet-18 | 10 | 82.8±0.54(**83.5**) |
| N-Caltech101 | Kim et al. (Kim & Panda, 2021) | SALT | VGG11 | 20 | 55.0 |
| | Li et al. (Li et al., 2022) | NDA | VGG11 | 10 | 78.2 |
| | Zhu et al. (Zhu et al., 2022) | TCJA-SNN† | VGGSNN | 14 | 78.5 |
| | **Our Method** | MTT | ResNet-18 | 10 | **81.74**±0.73(**82.32**) |

the experiment three times to report the mean and standard deviation. See appendix A.2 for training details.

On CIFAR10, our proposed technique outperforms InfLoR-SNN by 0.35% at T=6. In addition, our T=2 model performs similarly to their T=4 model. Our method also shows a 4.62% improvement on the standard ResNet-34 structure at T=6 and outperforms its SEW counterpart by 0.5 at T=4. For CIFAR10-DVS and N-Caltech101, our MTT method achieves an average accuracy of 82.8% and 81.45% and the best accuracy of 83.5% and 82.32%, respectively.

## 5.2 TEMPORAL FLEXIBILITY

In this section, we demonstrate the temporal flexibility of TFSNN and the advantages it brings. **Temporal Flexibility Across Settings** We first test TFSNN on the dataset and network structure mentioned in the last section (see Table 4). On static datasets, TFSNN performs fairly well at different time steps. On DVS datasets, though achieving temporal flexibility is more challenging due to the temporal characteristics, TFSNN can still be generalized to all time steps. We then further compare our method with recent ANN-SNN conversion methods in Table 6. Jiang et al. (2023) focuses on ultra-low-latency inference, and their results at T=1, and T=2 are the current SOTA in the field of ANN-SNN Conversion. Despite the well-designed fine-tuning procedure these SOTA conversion methods entail, MTT still outperforms them significantly at T=1 and T=2 while remaining comparable to them for higher time steps. Note that for a fair comparison, we adopt the same data augmentation policy as these methods.

**Table 4:** Accuracy of different inference time steps.

| Dataset | Model | T=2 | T=3 | T=4 | T=5 | T=6 |
|---|---|---|---|---|---|---|
| CIFAR100 | ResNet-19 | 80.35 | 81.14 | 81.51 | 81.73 | 81.98 |
| CIFAR10 | ResNet-19 | 96.20 | 96.62 | 96.75 | 96.76 | 96.84 |
| ImageNet | ResNet-34 | 65.23 | 67.58 | 67.54 | 68.02 | 68.34 |
| **DVS Dataset** | **Model** | **T=2** | **T=4** | **T=6** | **T=8** | **T=10** |
| CIFAR10-DVS | ResNet-18 | 72.47 | 79.9 | 81.0 | 82.0 | 82.8 |
| N-Caltech101 | ResNet-18 | 69.46 | 75.70 | 78.68 | 80.10 | 81.74 |

**Table 5:** Combine MTT with SEENN.

| Method | T=1.20 | T=1.09 |
|---|---|---|
| SEENN-I (Li et al., 2023b) | 96.38 | 96.07 |
| SEENN-I + MTT | **96.58** | **96.08** |

**Table 6:** Compare with SOTA ANN-SNN conversion methods on CIFAR100, ResNet18. $T_{max} = 6$ is used for MTT.

| Method | T=1 | T=2 | T=4 | T=8 | T=16 | T=32 | T=64 |
|---|---|---|---|---|---|---|---|
| QCFS (Bu et al., 2023) | - | 70.29 | 75.67 | 78.48 | **79.48** | **79.62** | **79.54** |
| SlipReLU (Jiang et al., 2023) | 71.51 | 73.91 | 74.89 | 75.40 | 75.41 | 75.30 | 74.98 |
| MTT | **72.09** | **76.54** | **78.47** | **78.90** | 79.17 | 79.25 | 79.42 |

**Table 7:** NMNIST Test Accuracy on Asynchronous Chip

| Method | GPU | Speck2e Devkit |
|---|---|---|
| SDT | 98.61 | 96.18 (-2.43) |
| MTT | 97.79 | **98.5 (+0.71)** |

**Combine with Dynamic Inference Time Step** We combine SEENN (Li et al., 2023b) with the MTT-trained ResNet19 on CIFAR10 to show how temporal flexibility benefits the confidence-based temporal dynamic method. We observed a performance boost under the same average inference time when compared with their reported results trained by TET, testifying the suitability of TFSNN to dynamic time step inference methods.

**Compatibility with Asynchronous Chip** Since the performance of TFSNN is largely decoupled from the influence of the time step due to the design of MTT and the concept of timestep also doesn't exist on asynchronous chips, TFSNN is naturally more suitable for deployment on asynchronous SNN chips. To demonstrate this, we deploy the MTT-trained SNN on Synsense's Speck2e Devkit (Richter et al., 2023) for asynchronous testing. Specifically, we train two networks on the NMNIST dataset using SDT and MTT, respectively, and then deploy them on the Speck2e Devkit, as detailed in Table 7.

## 5.3 ANALYSIS VALIDATION EXPERIMENTS

In this part, we conduct validation experiments. We first verify our theory of gradient and generalization in section 4.2, and then perform TFSNN accuracy estimation described in section 4.3.3.

**Alleviate Gradient Problem** According to our analysis in section 4.2, our method mitigates the effect of gradient noise on model training and thus can achieve better performance than SDT on deep networks. To verify this theory, we hand-make two deep networks, B-ResNet-50 and B-ResNet-72 (see appendix A.6 for details), and evaluate the performance of it trained for 100 epochs by MTT and SDT on the CIFAR10 dataset. The results are shown in Table 8. Compared to SDT,

**Table 8:** Training B-ResNet-50/72 on CIFAR10 dataset.

| Method | B-ResNet-50 | B-ResNet-72 |
|--------|-------------|-------------|
| SDT T=4 | 82.82 | 49.41 |
| MTT T=4 | **88.16** | **53.71** |

MTT achieved higher accuracy on B-ResNet-50 ($+5.34\%$) and B-ResNet-72 ($+4.3\%$). Both MTT and SDT are affected by gradient noise, but MTT effectively alleviates the performance loss caused by gradient errors.

**Generalization** As mentioned earlier in section 4.2, our method makes the network parameters sound against network structure changes like dropout and can generalize better. We further verified this through experiments with ResNet-18 on CIFAR100. A common method to measure the model's generalizability is noise injection. First, we randomly inject Gaussian noise $\mathcal{N}(0, \sigma^2)$ to weights, where $\sigma^2$ is the variance of the noise. For each $\sigma^2$, we run the experiment 5 times and plot the

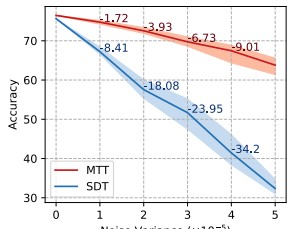 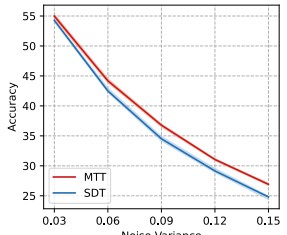

**Figure 5:** Accuracy of models with weights injected with Gaussian noise

**Figure 6:** Accuracy of models with inputs injected with Gaussian noise

mean, maximal, and minimal accuracy in Fig. 5. Results show that the weights trained by MTT are more robust against noise. Then, we inject Gaussian noise into the inputs instead, and also run the experiment 5 times each $\sigma^2$, the results are shown in Fig. 6. In addition, to solidify the conclusion, we also inspect the generalizability in other metrics (see appendix A.11). All the experiments indicate that the model obtained by MTT is more generalizable, and MTT has an effect similar to regularization.

**TFSNN Accuracy Estimation** Here, we validate our hypotheses in section 4.3.3. We randomly sample 18 kinds of TFSNN (different time step combinations) and obtain their test accuracy for solving the hypothesis equation in section 4.3.3. The weight parameters we obtained are $\{0.93, 0.53, 0.59, 0.67, 1.22, 0.48, 1.36, 0.18\}$, and the constant value $c$ is 67.11. This result supports our hypothesis 1), which suggests that all blocks' time step increment positively contributes to the accuracy of the final network. Some blocks, such as blocks 1, 5, and 7, have a greater contribution. Then, we resample 1000 TFSNN and validate their estimated accuracy and their test accuracy. The result (Fig. 7 (A)) shows that our method can effectively predict the actual testing accuracy of TFSNN. Finally, we use the spike frequency and accuracy estimation to build a combinatorial optimization equation for searching the optimal TFSNN (see appendix A.3 for detail). For example, by setting the energy cost that is lower than a default TFSNN (the time step of all blocks is 3), we discover the optimal combination of block time steps is $\{3, 2, 2, 3, 5, 3, 6, 2\}$. The selected TFSNN acquires an accuracy of 75.38%, which is 0.62% higher than the default.

## 5.4 ABLATION STUDY

**Evolution from NMT to MTT** Our improvement to NMT lies in dividing the network into many stages with the same time step and constructing a TFSNN, and NMT can be seen as a special case of the coarsest granularity of TFSNN division. In this section, we try different partitioning granularities, namely different numbers of blocks per stage $g$, to further validate the effect of adding more time structures to the optimization space. We train TFSNNs with g=1,2,4,8, and a single

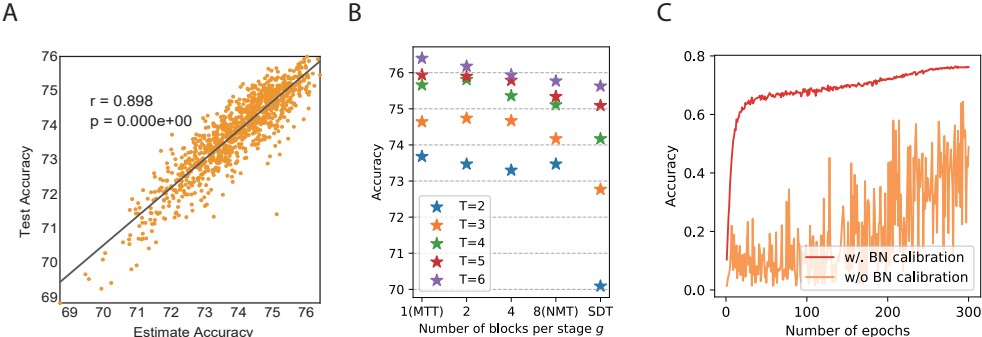

**Figure 7:** (A) Correlation curve between estimate accuracy and true accuracy. (B) Accuracy of ResNet-18 with different granularity where $g$=8 denotes NMT and $g$=1 denotes MTT. SDT denotes a single model trained at T=6 and tested across T=2,3,4,5. (C) Training ResNet-18 on CIFAR100 and tracing the test accuracy with and without BN calibration.

normal SNN with SDT. Then, we assess model accuracy for each with T=2,3,4,5,6. The results are shown in Fig. 7 (B). As expected, with $g$ continued to reduce and more temporal structures added to the optimization space, the overall performance is generally improved.

**Effectiveness of BN Calibration** In this section, we studied the necessity of BN calibration. We first trained two ResNet-18 on CIFAR100 and tracked their accuracy, one with BN calibration on 10 batches before testing and one simply using running means and variances. The results are shown in Fig. 7 (C). Our experiment indicates that without correct BN statistics, the model suffers huge accuracy degradation and that BN calibration effectively ameliorates the degradation.

## 6 CONCLUSION

In this work, we focus on training an SNN that can run at any simulation time with a set of fixed parameters. We start with a simple method, naive mixture training, and analyze the reason why NMT is effective. Based on this, we design the training method mixed time step training (MTT) to train a temporal flexible SNN (TFSNN) that can degrade to a normal SNN at any time step. We validate our theory with experiments and demonstrate the ability of TFSNN to generalize at different inference time steps. Although we found that TFSNN has the potential to outperform SNN with the same energy consumption when each block time step is different, we are unable to determine the best combination of T during training and need the validation set for analysis. Even so, we believe that this work is instructive for designing the structure of SNN.

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

# A APPENDIX

## A.1 EXPERIMENTS ON VGG STRUCTURES

We evaluated our method on the CIFAR100 dataset using VGG architectures, besides ResNets. We treated each layer as a stage in the VGG series. During experimentation, we observed that VGG16 with three fully connected (fc) layers could not be trained effectively using the standard direct training approach (its accuracy remained limited at 1%). To tackle this issue, we merged the last three fc layers of VGG16 into one and named the resulting architecture VGG14. We set $T_{max} = 5$, $s = 3$, and computed the mean and standard deviation of three runs. We used the SGD optimizer to train the model, with a learning rate of 0.1, a weight decay of 0.0005, and a batch size of 256. The results, presented in Table 9, indicate the effectiveness of our approach on VGG structures.

**Table 9:** Accuracy of VGG on CIFAR100

| Methods | Model | T=2 | T=3 | T=4 | T=5 |
|---------|-------|-----|-----|-----|-----|
| MTT | VGG14 | 73.53±0.09 | 74.52±0.07 | 75.27±0.14 | 75.72±0.10 |
| InfLoR-SNN | VGG16 | - | - | - | 71.56±0.10 |

## A.2 TRAINING DETAILS

**CIFAR** The CIFAR10/CIFAR100 dataset comprises 50K training images and 10K test images with a 32×32 pixel resolution. For CIFAR100, we train a ResNet-19 TFSNN using the MTT pipeline for 300 epochs with a batch size of 256 and a $T_{max}$ of 6. Following the practice in GLIF (Yao et al., 2022), the last 2 fully connected layers of ResNet-19 are replaced with a single fully connected layer. We employ the SGD optimizer with a weight decay of 0.0005 and a learning rate of 0.1 cosine decayed to 0. To make a fair comparison with the state-of-the-art (SOTA) work (Li et al., 2021b; Guo et al., 2022; Yao et al., 2022), AutoAugment (Cubuk et al., 2018) and Cutout (DeVries & Taylor, 2017) are applied to both CIFAR10 and CIFAR100 datasets. However, these augmentation techniques are only used for comparative experiments and temporal flexibility experiments, and not for other experiments.

**ImageNet** ImageNet (Deng et al., 2009) contains more than 1280k training images and 50k test images. We use the standard data processing flow to crop each image to a size of 224×224. We deploy the ResNet-34 structure, however, with the removal of the first max-pooling layer and changing the stride of the first basic block from 1 to 2 (Zheng et al., 2021; Yao et al., 2022). We train the model for 160 epochs with a batch size of 512 and a $T_{max}$ of 6. We utilize the AdamW optimizer with a weight decay of 0.02 and a learning rate of 0.004 cosine decayed to 0.

**DVS-Dataset** CIFAR10-DVS and N-Caltech101 are neuromorphic datasets widely used in SNN experimentation. We divide the dataset into a 9:1 ratio and merge all events to form ten frames, which, similar to previous work (Li et al., 2021b; Deng et al., 2022), we resize to 48×48. For both these datasets, we adopt a random horizontal flip and rotate the frames up to 5 pixels as augmentation techniques. We employ the additional temporal inversion policy (Shen et al., 2023) uniquely for N-Caltech101. For these datasets, we use a $T_{max}$ of 10, a batch size of 50, and train the TFSNN ResNet-18 model for 300 epochs. The optimizer we choose is SGD, with a weight decay of 0.0005, and a learning rate of 0.1, which we cosine decay to 0. While training the DVS dataset, we take only the first $t$ frames of the ten frames where $t$ denotes the time step of the input stage, to feed into the network.

## A.3 DETAILS OF COMBINATORIAL OPTIMIZATION

As previously mentioned, the accuracy formula of TFSNN is given by the expression $\sum_{i=1}^{I} K_i \sqrt{\log_2 t_i} + c$, where $K_i$ represents the contribution of each block, $I$ is the number of blocks, and $c$ is a bias. We randomly select 18 distinct temporal configurations ($\boldsymbol{t}$) and evaluate their accuracies on the test set, resulting in 18 pairs of temporal configurations and their corresponding accuracies. Using the least squares method, we compute the values of $K_i$ and $c$ from the collected data. Next, we estimate the average firing rate ($R_i$) of each block in a unified SNN of T=6. Then, the energy consumption of a specified temporal configuration $t$ can be approximated as $\sum_{i=1}^{I} t_i \cdot R_i$. For example, the estimated energy consumption of a unified SNN with T=4 is calculated as $EC_4 = \sum_{i=1}^{I} 4R_i$. Based on this, we can obtain a group of TFSNNs with lower

energy consumption ($EC$) for a given T=$T_g$, and we aim to identify the TFSNN with the maximum estimated accuracy from this set. This is formulated as the following optimization problem:

$$\text{maximize} \quad \text{ACC}_{\text{estimated}} = \sum_{i=1}^{I} K_i \sqrt{\log_2 t_i} + c$$

$$\text{s.t.} \quad \sum_{i=1}^{I} t_i \cdot R_i \leq EC_{T_g}$$

$$T_{min} \leq t_1, t_2, \ldots, t_l \leq T_{max},$$

where $t_i$ is the $i$-th component of temporal configurations $t$ and $EC_{T_g}$ is the given uper bound of the TFSNN energy.

In order to solve the above problem, we adopt the depth-first search (DFS) algorithm to search in the solution space. To obtain a more accurate accuracy for each temporal configuration $t$, we perform three times of BN calibrations and take the average of the accuracies.

## A.4 DESIGN TTM GROUPING POLICY

We previously stated our policy's objective is to partition $l$ frames into $k$ groups ($l \geq k$) as evenly as possible. In this section, we mathematically interpret the design. We will start by describing the grouping process in a different manner. The $l$ frames are viewed as $l$ adjacent intervals of length 1 over the rational number domain with the $i$-th frame starting at $i-1$ and ending at $i$. We define $c_i$ as the boundary between group $i$ and group $i-1$. Here, $i$ ranges from 1 to $k$, and $c_1$ is 0. Ideally, $c_i = (i-1) \cdot l/k$ is set to group frames most evenly. Nevertheless, this strategy produces non-integer $c_i$, which results in atomic frames' division when $l$ is not a multiple of $k$. To solve this problem, we retreat and set $c_i$ to the nearest integer and get

$$c_i = \lfloor \frac{(i-1) \cdot l}{k} - \varepsilon \rceil, \tag{11}$$

where $\varepsilon$ is a small constant used to determine $c_i$ when the distances to the closest two integers are equal. As $b_i$ must be the frame directly following the boundary $c_i$ ($b_i = c_i + 1$), we obtain Eq. 8.

## A.5 DERIVATION OF THE BACKPROPAGATION FORMULA FOR LIF

In this section, we derive Eq. 7 from the forwarding formula. We derive $\partial u(t)/\partial v(t)$ from Eq. 4 first:

$$\frac{\partial u(t)}{\partial v(t)} = 1 - s(t) - v(t) \cdot \frac{\partial s(t)}{\partial v(t)}. \tag{12}$$

Then, we consider the derivation of $\partial u(t)/\partial v(t-1)$. According to Fig. 2, $\partial u(t)/\partial v(t-1)$ can be calculated as follows

$$\frac{\partial u(t)}{\partial v(t-1)} = \frac{\partial u(t)}{\partial v(t)} \frac{\partial v(t)}{\partial u(t-1)} \frac{\partial u(t-1)}{\partial v(t-1)} = \tau \frac{\partial u(t)}{\partial v(t)} \frac{\partial u(t-1)}{\partial v(t-1)}. \tag{13}$$

By combining multiple Eq. 13, we get

$$\frac{\partial u(t)}{\partial v(t-n)} = \tau^n \prod_{i=t-n}^{t} \frac{\partial u(i)}{\partial v(i)}. \tag{14}$$

Finally, we get the complete expression for $\partial s(t)/\partial I(t-n)$ as follows

$$\frac{\partial s(t)}{\partial I(t-n)} = \frac{\partial s(t)}{\partial v(t)} \frac{\partial v(t)}{\partial u(t-1)} \frac{\partial u(t-1)}{\partial v(t-n)} \frac{\partial v(t-n)}{\partial I(t-n)}$$

$$= \frac{\partial s(t)}{\partial v(t)} \cdot \tau^n \prod_{i=t-n}^{t-1} [(1-s(i)) - v(i) \cdot \frac{\partial s(i)}{\partial v(i)}] \tag{15}$$

## A.6 DETAILS OF HANDMADE MODEL STRUCTURES

To demonstrate MTT's capacity for training deep networks, we created two deep networks, B-ResNet-50 and B-ResNet-72. We will present their specific structures in this section. Table 10 displays the architectures of these two networks. For convenience, we still use basic blocks for each block, and simply reconfigure the number of basic blocks in each part. Since the model is large, we use $T_{max} = 4$.

Table 10: Structures of our handmade deep networks on CIFAR.

| Stage | Output Size | B-ResNet-50 | | B-ResNet-72 | |
|---|---|---|---|---|---|
| conv1 | 32×32 | 3x3, 64 | | | |
| conv2_x | 32×32 | 3x3, 64 
 3x3, 64 | * 3 | 3x3, 64 
 3x3, 64 | * 3 |
| conv3_x | 16×16 | 3x3, 128 
 3x3, 128 | * 6 | 3x3, 128 
 3x3, 128 | * 8 |
| conv4_x | 8×8 | 3x3, 256 
 3x3, 256 | * 9 | 3x3, 256 
 3x3, 256 | * 16 |
| conv5_x | 4×4 | 3x3, 512 
 3x3, 512 | * 6 | 3x3, 512 
 3x3, 512 | * 8 |
| FC | 1×1 | average pool, fc, softmax | | | |

## A.7 $T = T_{MAX}$ BN STATISTICS VS. RECALCULATED BN STATISTICS

As previously mentioned, we discovered that the BN statistics of the $T=T_{max}$ network can be applied to other TFSNN with uniform T across blocks. In this section, we provide experimental verification for this observation. Our experiment involves testing the accuracy of one of our trained ResNet-19 models with two distinct BN layer information approaches. The first approach utilizes the statistics of the $T=T_{max}$ network, while the second approach recalculates the BN statistics individually for each time step T. For the latter approach, we calibrate the BN layer three times and report the average accuracy. Our experimental results demonstrated in Table 12 show that the mean and variance calculated at $T=T_{max}$ is applicable directly to other T values. Therefore, we can utilize the BN statistics of $T=T_{max}$ for other T directly, which saves the time required to calibrate the BN layers for other T values.

Table 11: Accuracy given sampling number $s$ and training epochs $e$.

| Sampling Num | $e \times s$ | | | |
|---|---|---|---|---|
| | 300 | 600 | 900 | 1200 |
| $s = 1$ | **75.61** | 76.13 | 76.23 | 75.78 |
| $s = 2$ | 75.53 | 76.37 | 76.31 | 76.36 |
| $s = 3$ | 75.25 | **76.45** | **76.47** | **76.44** |
| $s = 4$ | 74.81 | 75.74 | 76.41 | 76.42 |

Table 12: Test accuracy of a single model with two kinds of BN statistics.

| Method | TimeStep | | | |
|---|---|---|---|---|
| | 2 | 3 | 4 | 5 |
| $T=T_{max}$ stat | 80.21 | 81.06 | **81.51** | 81.82 |
| Recalculated stat | 80.21 | **81.23** | 81.44 | **81.85** |

## A.8 IMPACT OF DIFFERENT SAMPLING NUMBER AND TRAINING EPOCHS

In most previous experiments, we employed sampling number $s = 3$. In this section, we experiment with varying values of $s$, assess their effects at different epochs, and explain why we chose $s = 3$. We train ResNet-18s with $T_{max} = 6$ on CIFAR100 with varied $s$ and list their test accuracy at T=6. The results are displayed in Table 11. When $e \times s$ is constrained, the model requires more epochs to converge, necessitating a lower $s$. However, if trained across a sufficient number of epochs, sampling of $s$ structurally diverse networks can smooth the optimization of network parameters and improve performance. Specifically, we find $s = 3$ performs well and adopt $s = 3$ for most of the experiments.

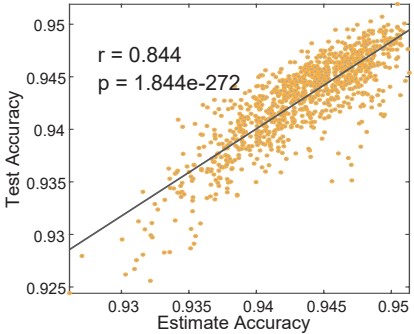

**Figure 8:** Correlation curve for estimate accuracy and test accuracy on CIFAR10

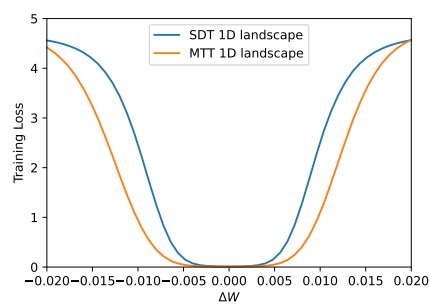

**Figure 9:** The 1D landscapes of ResNet18 trained by SDT and MTT on CIFAR100.

### A.9 TFSNN Accuracy Estimation on CIFAR10

In addition to CIFAR100, we also conducted experiments with ResNet-18 on CIFAR10. We randomly sample 18 temporal configurations as usual and solve the equation in appendix A.3. The weight parameters are $\{0.0049, 0.0028, 0.0017, 0.0037, 0.0037, 0.0004, 0.0005, 0.0017\}$, and the constant value $c$ is 92.13. Notes that block 1,4,5 have a higher contribution, and the 1,5 blocks are also highly weighted on CIFAR100, which may imply that the weights are partly related to the network structure. We then resample 1000 TFSNN and plot their estimate accuracy and test accuracy on CIFAR10 in Fig. 8. We also use the aforementioned combinatorial optimization strategy to search the optimal TFSNN under the energy consumption of T=3 and find $\{4, 2, 2, 3, 5, 2, 2, 4\}$, which achieves an accuracy of 94.70%, 0.21% higher than its T=3 counterpart.

### A.10 Loss Landscapes of MTT and SDT

To visually confirm the flatter minimum achieved by the model trained with MTT, we trained ResNet18 using SDT and MTT on CIFAR100 and plotted their loss landscapes in Fig. 9. We observed that MTT led the model to a flatter minimum which indicates improved generalizability.

### A.11 Verifying Generalizability Through Gradient Metrics

Apart from noise injection, another famous metric that indicates the generalizability is the length of the gradient on weights $||\frac{\partial \mathcal{L}}{\partial W}||$ and the inputs $||\frac{\partial \mathcal{L}}{\partial x_i}||$. For $||\frac{\partial \mathcal{L}}{\partial W}||$, we evaluate the length of the gradient of loss over the entire training set for the convolution layers. For $||\frac{\partial \mathcal{L}}{\partial x_i}||$, we calculate the mean value of the length of each input gradient. The model trained by MTT exhibits a shorter gradient of both weights and inputs (see Table 13), which implies the model's strong robustness and generalizability.

**Table 13:** The gradient statistics of the model trained by SDT and MTT.

| Methods | $\|\|\frac{\partial \mathcal{L}}{\partial W}\|\|$ | $\|\|\frac{\partial \mathcal{L}}{\partial x_i}\|\|$ |
|---|---|---|
| MTT | 11.59 | 1.81 |
| SDT | 38.08 | 7.78 |

**Table 14:** Results on seqMNIST.

| Methods | Acc |
|---|---|
| Our RNN | 56.22 |
| SNN SDT | 55.75 |
| SNN MTT | **64.56** |

**Table 15:** Results on Spiking Heidelberg Digits

| Methods | Acc |
|---|---|
| l=3 Repro by code of Hammouamri et al. (2023) | 75.26 |
| l=3 our SDT | 74.43 |
| l=3 our MTT | **79.68** |

**Table 16:** Results on Spiking Speech Commands

| Methods | Acc |
|---|---|
| Our SDT l=3 | 57.75 |
| Our MTT l=3 | **60.15** |

### A.12 EXPERIMENTS ON AUDIO AND SEQUENTIAL DATASETS

Our research reveals that TFSNN can function effectively as a time encoder when the temporal configuration vectors used for training are monotonically non-increasing.

To illustrate this adaptability, we present the performance of TFSNN on three distinct temporal tasks: seqMNIST, Spiking Heidelberg Digits and Spiking Speech Commands.

For the sequential task seqMNIST, we utilized a simple fc LIF SNN with 2 hidden layers of width 64 and set the time constant $\tau$ to 0.99. We also trained an RNN with 2 hidden layers of width 64 for comparison. The results are as shown in Table 14.

For the Spiking Heidelberg Digits, we adopt the plain 3-layer feed-forward SNN architecture proposed by Cramer et al. (2020), a fully connected SNN with an input width of 70 and 128 LIF neurons in each of the 3 hidden layers. The timestep of the first layer is fixed to the input timestep, while the timesteps of subsequent layers are restricted to monotonically non-increasing. We set $\tau = 0.9753$, which is equivalent to the parameter $\lambda$ in the work of Cramer et al. (2020), namely $1 - 1/\tau$ in most other articles, and train the model for 150 epochs. To ensure the validity of our results, we also reproduce the result using the code provided by Hammouamri et al. (2023). The results are as shown in Table 15.

Spiking Speech Commands (SSC) (Cramer et al., 2020) is a spiking dataset converted from Google Speech Commands v0.2 and is tailored for SNN. For SSC, we continue using the same architecture and the same parameters as we used in SHD, except that here we only train the model for 60 epochs. The results are as shown in Table 16.

