# OpenReview forum: "Temporal Flexibility in Spiking Neural Networks: A Novel Training Method for Enhanced Generalization Across Time Steps"
_ICLR.cc/2024/Conference — Submitted to ICLR 2024_

### Official Review · Reviewer_1Uof · 2023-10-26

**Soundness:** 3 good
**Presentation:** 2 fair
**Contribution:** 2 fair
**Rating:** 3
**Confidence:** 5

**Summary:**

This work aims to extend SNN learning adaptively to various time stamps. The proposed model is based on the discrete LIF neuron model and conduct the multi-step ResNet architectures in serial.

**Strengths:**

1. This paper provides lots of vivid plots for illustrating the workflow and topology of the proposed models, which is beneficial the understanding of readers.

2. The experimental results seem to be convincing. I have no doubts about the reproducibility of the experimental results.

**Weaknesses:**

1. This paper is hard to follow due to the poor organization and presence. For instance, it would be helpful to clearly state the purpose of the experiments and analyses in Subsections 4.2 and 4.3.

2. The language used in the paper is quite loose, leading to some confusion. For example, when the paper states that "Increasing the time step (T) enhances the SNN’s expressiveness," it would be beneficial to provide a precise description of "expressiveness." Additionally, the claim that "increasing the time step leads to an increase in the calculated surrogate gradient error" requires further explanation. The treatment of time steps in this work deviates from typical learning methodologies for SNNs, potentially causing confusion for readers, as $T$ here does not represent the length of a spike sequence.

2.1. The authors blur this concept here, making it difficult for readers to distinguish existing work from traditional practices.Clearly, the authors seem to aim to bolster the significance of their paper with this assertion. However, it must be acknowledged that this approach is not in line with standard practices, particularly for SNNs. When employing this method, SNNs may not showcase the capability to outperform conventional neural networks. Instead, their representational capacity may be limited to a subset of RNNs.

3. The title of the paper may be considered somewhat overstated. The claim of "temporal generalization" lacks substantial support throughout the paper. The assertion of theory verification in Section 5.3 requires clarification.

4. The temporal transformation module introduced in the paper bears similarity to existing concepts, such as multi-step RNNs. Providing a clear distinction or novelty in this module would enhance the paper's contribution.

**Questions:**

As mentioned above.

---

> ### Author Response · Authors · 2023-11-21
> **Response to Reviewer 1Uof (Part 1)**
>
> **Comment 1: This paper is hard to follow due to the poor organization and presence. For instance, it would be helpful to clearly state the purpose of the experiments and analyses in Subsections 4.2 and 4.3.**
>
> Thank you for your kind reminder. We have added a summary at the beginning of Section 4 to provide a clear description of the purpose of all the experiments and analyses.
>
> "In this section, we introduce our method Mixed Timestep Training (MTT), and its derivation process. We first experimented with a simple method, Naive Mixture Training (NMT), in Section 4.1 to bring up the temporal flexibility of SNNs. As a result, NMT not only succeeded in enabling SNNs to generalize across different time steps but also consistently outperformed the SDT-trained model at every single time step (even at $\mathrm{T}=\mathrm{T}_{max}$). In Section 4.2, we delve into the reasons behind NMT's performance improvement, examining both gradient and generalizability. Subsequently, in Section 4.3, we present our final method, MTT—an enhanced version of NMT that produces Temporal Flexible SNN (TFSNN) with high adaptiveness to time steps."
>
> In addition to this, we have optimized the presentation and organization of the whole article. The texts highlighted in blue in our new version of the paper are our changes.
>
> ----------------
>
> **Comment 2: The language used in the paper is quite loose, leading to some confusion. For example, when the paper states that "Increasing the time step (T) enhances the SNN’s expressiveness," it would be beneficial to provide a precise description of "expressiveness." Additionally, the claim that "increasing the time step leads to an increase in the calculated surrogate gradient error" requires further explanation. The treatment of time steps in this work deviates from typical learning methodologies for SNNs, potentially confusing readers, as T here does not represent the length of a spike sequence.**
>
> Thanks for your suggestion. We agree that we should add more explanation to avoid the confusion for reader. Therefore, we add additional descriptions of "expressiveness", "gradient error", and other related concepts in the revised manuscript. Hope our revision makes the context more clearly.
>
> We included the following sentences in Section 4:
>
> "expressiveness" refers to the information encoding capability of the Spike trains. Due to the binary nature of the spike-based inter-layer information transmission in SNNs, their information encoding capacity is not as strong as that of ANNs. For example, a spike train composed of three spike values (each value can be 0 or 1) can only express 2^3 possible values. while the information encoding capacity of high-precision numerical data is almost unlimited (2^32). Fortunately, spike trains (SNN layer output) have both spatial and temporal dimensions, and expanding these dimensions can enhance the total information encoding capacity, i.e., "expressiveness". The spatial dimension is constrained by the network structure, while the temporal dimension is controlled by the SNN time step T. Increasing the time step T can lead to an improvement in the information encoding capacity, thereby enhancing the “expressiveness” of SNN.
>
> "gradient accuracy" refers to the precision of the parameter gradients obtained by backpropagation. The reason for this notion arises from the fact that in the direct training of SNNs, the gradients computed by the backpropagation algorithm contain errors. These errors originate from the term $\frac{\partial s(i)}{\partial v(i)}$ in Eqn. 7, where this term is calculated as an approximated value by surrogate gradient during backpropagation because its original function is a step function. And there exists a certain error between the approximated value and the true value. As shown in Eqn. 7, as the time steps T increases, there are more instances of $\frac{\partial s(i)}{\partial v(i)}$, leading to a larger error in the calculated parameter gradient. [1] try to quantify the gradient error by the similarity between the gradient obtained through backpropagation (Eqn. 7) and the gradient computed by the finite differences (FDG).

---

> ### Author Response · Authors · 2023-11-21
> **Response to Reviewer 1Uof (Part 2)**
>
> **Comment 2.1: The authors blur this concept here, making it difficult for readers to distinguish existing work from traditional practices. The authors seem to aim to bolster the significance of their paper with this assertion ... not in line with standard practices ... Instead, their representational capacity may be limited to a subset of RNNs.**
>
> Thanks for mentioning these two valuable points. Due to the scope of the current work, we didn't discuss the meaning of time steps in SNN/TFSNN and its difference from the traditional practice of SNN setups. Regarding your concerns, we agree that we should add clarifications here.
>
> First want to state that we did not intend to blur any concepts here. The setup of using T as the latency of the network is a widely acceptable setup in many existing works, such as [3, 4, 5, 6, 7, 8, 9]. Differentiating the operation time steps and spike sequence length allows SNN to deal with non-spike signals and extends the applicable fields of spiking neural networks.
>
> Second, regarding our contribution and its feasibility with SNN, we understand that different setups would lead to different requirements for the deployment. For fully asynchronous chips like Speck, the calculation process is almost the same as biological neurons where each of the neurons receives and releases spikes independently. Therefore, the concept of the time step doesn't exist on this kind of chip and the time step introduced during GPU training is merely for a discrete simulation of the real asynchronous on-chip scenario. For synchronous/semi-asynchronous chips, e.g. Loihi, the concept of time step does exist in the latter type of chip, which usually needs to be synchronized with a clock. The on-chip model on this kind of chip and the GPU model are theoretically the same.
> Regarding the datasets commonly used for SNN benchmarking, there are three typical families, traditional static datasets (e.g. CIFAR10, CIFAR100, ImageNet), traditional temporal datasets (e.g. Google Speech Command), and event datasets (e.g. DVS-CIFAR10, N-MNIST, SHD). The event datasets are either converted from traditional ones or recorded by spiking sensors such as DVS cameras. Each input of an event dataset is an event stream consisting of tuples as $(t_i, x_i)$ where $t_i$ is the emission time of the event and $x_i$ is the input position of the event.
>
> Now, we demonstrate the advantages of our method under 3 common settings respectively.
> 1. Deployment on synchronous/semi-asynchronous chips with static datasets (e.g. CIFAR) is the most frequently discussed topic. An input from static datasets is repeated for $T$ times and then fed into an SNN operating at time step $T$. This is also an essential prerequisite for ANN-SNN conversion and dynamic inference time steps. Here, the significance of TFSNN lies in the training-free energy-performance balance and the performance boosts it brings when taking MTT as a training method.
> 2. For synchronous/semi-asynchronous chips with temporal and event datasets (e.g. DVS-CIFAR10), the significance of TFSNN mainly lies in the performance boosts, with possible energy-performance balance on a few number of datasets (e.g. DVS datasets shown in the paper). Inputs from temporal and event datasets, unlike static ones, are first collected into $T$ time frames with the same length and then fed into the network. We found that MTT can provide SNNs operating at $\mathrm{T}_{max}$ with higher performance on temporal and sequential tasks after ensuring a non-increasing time step sequence relative to input distance during temporal configuration sampling in the training phase.
> 3. The asynchronous chips are usually connected after a spiking sensor (e.g. a DVS camera) and are often used for event datasets. The asynchronous chips are also where the real power of SNN is exhibited. When training SNNs on GPU, the event inputs are collected into $T$ time frames for simulating chips discretely. After training, the weights of the model are downloaded to the chip directly for asynchronous inference. While existing direct training methods are only for a specific $T$, our TFSNN is largely disentangled from the influence of $T$ and thus suffers little performance loss after being deployed (see our response to reviewer wYLe).
>
> Third, we have to clarify the performance of existing SNNs can hardly surpass their ANN counterparts. The real advantage of SNN is not its performance but its low latency and low energy consumption when running on neuromorphic chips. This mechanism inspired by biological neurons allows SNNs to operate at extremely low power. For example, the authors of [1] calculated the energy consumption for various SNNs. Notably, for a Spiking ResNet-101 operating at T=4, the dynamic energy consumption of a single $224\times 224$ ImageNet input is only 4.53mJ. In [2], the authors deployed their SNNs on Loihi and reported a running power of 0.81~0.82W in Table 4, which is about 50 times more efficient than GPU models.

---

> ### Author Response · Authors · 2023-11-21
> **Response to Reviewer 1Uof (Part 3)**
>
> **Comment 3: The title of the paper may be considered somewhat overstated. The claim of "temporal generalization" lacks substantial support throughout the paper. The assertion of theory verification in Section 5.3 requires clarification.**
>
> Thank you for your questions, we would like to answer these two questions separately.
>
> I. Support for the claim of "temporal generalization"
> Although many of our experiments are related to theory validation and further exploration of our method, we provided multiple experiments to support the temporal generalizability achieved by our method. We designed Section 5.2 for this validation, where we first directly tested the performance of TFSNN at different time steps across different datasets. The related results can be seen in Table 4. We also demonstrate our method's compatibility with dynamic inference time steps methods (e.g. SEENN) in Table 5 to indirectly show the temporal flexibility of our method.
> The most decisive result comes when we compare our method with ANN-SNN conversion SOTA in Table 6. Conversion is a well-known method that exhibits temporal flexibility at high timesteps (e.g. $\mathrm{T}\ge 64$) but is only suitable for static datasets (since they have to train an ANN with identical structure first) and fails to generalize across low time steps. We find our TFSNN not only exhibits competitive performance at high timesteps similar to conversion methods but also generalizes well under extremely low timesteps. In short, we highlight the strengths of our method in that
> - Our method generalizes well across both high and low timesteps.
> - Existing SOTA conversion-based methods require fine-tuning or time-steps-related ANN redesigning for a specific time step, while our TFSNN is a zero-shot method and doesn't need any extra fine-tuning when switching to a different timestep.
> - Conversion methods are only applicable to static datasets, while our MTT method can handle both static, temporal, and event datasets.
>
> II. The lack of clarification in Section 5.3
> We appreciate your suggestion concerning Section 5.3. The original version indeed is somewhat confusing due to the excessively simple summary in the beginning and the lack of back-reference to former sections in the body part. We modified the summary to make it clearer and added essential references to the related part in the former sections.
>
> A new summary at the beginning of Section 5.3 of the latest version: "In this part, we conduct validation experiments. We first verify our theory of gradient and generalization in section 4.2, and then perform TFSNN accuracy estimation described in section 4.3.3."

---

> ### Author Response · Authors · 2023-11-21
> **Response to Reviewer 1Uof (Part 4)**
>
> **Comment 4: The temporal transformation module introduced in the paper bears similarity to existing concepts, such as multi-step RNNs. Providing a clear distinction or novelty in this module would enhance the paper's contribution.**
>
> Thank you for your valuable advice! If we understand correctly, multi-step RNN is the derivative of the concept of single-step RNN. A normal single-step RNN first receives an input $x_t$ at the beginning of each time step, then updates its hidden state $h_t$, and finally outputs $y_t$ at the end of the time step. A multi-step RNN (e.g. $k$ steps), however, consecutively receives inputs and updates the state for $k$ time steps before finally producing the output.
>
> We first distinguish between RNNs and SNNs. We understand the similarity between the GPU implementation of SNN and RNN, since SNN also receives a sequence of input at the beginning, then updates its membrane potential (similar to the hidden state), and eventually produces binary outputs at the end of each time step. Nevertheless, we still have to mention the obvious structural differences between an RNN layer and an SNN layer. First, after receiving the inputs, RNNs update their hidden state by a recurrent fully connected layer while the membrane potential update of every spiking neuron is independent of each other. Second, when stacking multiple RNN/SNN layers together, an RNN layer will directly receive the output (hidden states) of the former layer as the input, but two layers of SNN neurons are often connected with a fully connected layer or convolution layer.
>
> Next, we compare the difference between multi-step RNN and our proposed TTM. There are three major differences.
> - A multi-step RNN has hidden states while our TTM doesn't have any parameters or hidden states and only serves as a plug-in that groups and merges adjacent time steps.
> - A multi-step RNN is equivalent to a single-step RNN when dropping the first $k-1$ outputs within every $k$ time step, while our TTM merges all the outputs within the same time step group.
> - A multi-step RNN can only produce an output sequence shorter than the input, while our TTM can perform the transformation from a short sequence to a longer one.
>
> We hope our response has answered your questions and look forward to hearing from you if you have further concerns!
>
> -------------------------------
>
> [1] Chen, Guangyao, et al. "Training Full Spike Neural Networks via Auxiliary Accumulation Pathway." arXiv preprint arXiv:2301.11929 (2023).
>
> [2] Stewart, Kenneth Michael, et al. "Speech2Spikes: Efficient Audio Encoding Pipeline for Real-time Neuromorphic Systems." Proceedings of the 2023 Annual Neuro-Inspired Computational Elements Conference. 2023.
>
> [3] Fang, Wei, et al. "Deep residual learning in spiking neural networks." Advances in Neural Information Processing Systems 34 (2021): 21056-21069.
>
> [4] Rathi, Nitin, and Kaushik Roy. "Diet-snn: Direct input encoding with leakage and threshold optimization in deep spiking neural networks." arXiv preprint arXiv:2008.03658 (2020).
>
> [5] Che, Kaiwei, et al. "Differentiable hierarchical and surrogate gradient search for spiking neural networks." Advances in Neural Information Processing Systems 35 (2022): 24975-24990.
>
> [6] Kim, Youngeun, et al. "Exploring temporal information dynamics in spiking neural networks." Proceedings of the AAAI Conference on Artificial Intelligence. Vol. 37. No. 7. 2023.
>
> [7] Yan, Zhanglu, Jun Zhou, and Weng-Fai Wong. "Near lossless transfer learning for spiking neural networks." Proceedings of the AAAI conference on artificial intelligence. Vol. 35. No. 12. 2021.
>
> [8] Meng, Qingyan, et al. "Training much deeper spiking neural networks with a small number of time-steps." Neural Networks 153 (2022): 254-268.
>
> [9] Li, Yuhang, et al. "Input-Aware Dynamic Timestep Spiking Neural Networks for Efficient In-Memory Computing." arXiv preprint arXiv:2305.17346 (2023).

---

> ### Author Response · Authors · 2023-11-23
>
> As the discussion deadline is drawing near, could you please consider reviewing our response and reconsidering our work, if convenient for you? We're open to any additional questions and would be delighted to provide further clarification. It would be wonderful if we could address any concerns you may have. Thank you for your time and valuable feedback!

---

### Official Review · Reviewer_wYLe · 2023-10-30

**Soundness:** 3 good
**Presentation:** 2 fair
**Contribution:** 3 good
**Rating:** 6
**Confidence:** 4

**Summary:**

This paper proposes temporal flexible spiking neural networks (TFSNN) and mixed time step training (MTT) method to improve the performance of SNNs under different time steps. The paper first demonstrates the effectiveness of naïve mixture training with different time steps during training, and then, inspired by this, proposes TFSNN with different time steps for each block and MTT training method. Experiments on static and neuromorphic datasets demonstrate superior performance and temporal flexibility of the proposed method, as well as the potential to discover optimal combination of block time steps with energy constraint.

**Strengths:**

1. The paper proposes a new method to achieve superior experimental results on large-scale datasets. The paper also conducts extensive analysis experiments.

2. The idea of allocating different time steps to different blocks is interesting. It may adaptively allocate energy based on the contributions of different blocks and make a balanced combination.

**Weaknesses:**

1. The neuromorphic hardware may not support different time steps for different blocks. For the discovered optimal combination of block time steps {3,2,2,3,5,3,6,2}, it may not be deployed for the real energy efficiency of SNNs. There can also be more discussion on the compatibility of the layer-wise adaption of time steps and the thought of (asynchronous) parallelism for neuromorphic computing.

2. Theoretical analysis is limited and there are many informal claims. For example, in Section 4.2, “we believe … balance between SNN expressiveness and gradient accuracy …” --- what is the formal definition of “expressiveness” and “gradient accuracy” and what’s the quantitative measurement of them? And “the new training loss may not converge, leading SNN to jump out of the local minimum point” --- to which loss the “local minimum” refers and why not converging can certainly lead to “jump out of the local minimum point” rather than move surrounding it? Is there any formal theoretical definition and analysis for these claims?

**Questions:**

See weakness.

---

> ### Author Response · Authors · 2023-11-21
> **Response to Reviewer wYLe (Part 1)**
>
> **Comment 1: The neuromorphic hardware may not support different time steps for different blocks. For the discovered optimal combination of block time steps {3,2,2,3,5,3,6,2}, it may not be deployed for the real energy efficiency of SNNs. There can also be more discussion on the compatibility of the layer-wise adaption of time steps and the thought of (asynchronous) parallelism for neuromorphic computing.**
>
> Thank you for your feedback. Regrettably, current synchronous devices do not facilitate mixed timestep inference. Consequently, the identified combination cannot be deployed directly; we utilized it solely for exploratory purposes to gain valuable insights. However, since the performance of TFSNN is largely decoupled from the influence of the time step due to the design of MTT and the concept of timestep also doesn't exist on asynchronous chips, TFSNN is naturally more suitable for deployment on asynchronous SNN chips. To demonstrate this capability, we deployed the MTT-trained SNN on Synsense's Speck2e Devkit for asynchronous testing. Specifically, we trained two networks on the NMNIST dataset using SDT and MTT, respectively, and then deployed them on the Speck2e Devkit, as detailed in Table R1.
>
> **Table R1 Test Accuracy on NMNIST(Adamw,lr=1e-3,batch_size=16,epochs=10)**
> |Methods|GPU| Speck2e Devkit|
> |-|-|:-:|
> |SDT|98.61|96.18|
> |MTT|97.79|98.5|
>
> As a result, the SDT-trained model suffered a performance drop of about 2\% while the MTT-trained model maintained and even achieved a higher performance on Speck2e Devkit.

---

> ### Author Response · Authors · 2023-11-21
> **Response to Reviewer wYLe (Part 2)**
>
> **Comment 2: Theoretical analysis is limited and there are many informal claims. For example, in Section 4.2, “we believe … balance between SNN expressiveness and gradient accuracy …” --- what is the formal definition of “expressiveness” and “gradient accuracy” and what’s the quantitative measurement of them? And “the new training loss may not converge, leading SNN to jump out of the local minimum point” --- to which loss the “local minimum” refers and why not converging can certainly lead to “jump out of the local minimum point” rather than move surrounding it? Is there any formal theoretical definition and analysis for these claims?**
>
> Thanks for your suggestion, indeed we have not provided clear definitions for the words “expressiveness” and “gradient accuracy” in the manuscripts. In the new version, we have included the following sentence to prove definitions for the two words (See the blue highlighted text in Sec 4.2 of the revised version):
>
> “expressiveness” refers to the information encoding capability of the spike trains. Due to the binary nature of the spike-based inter-layer information transmission in SNNs, their information encoding capacity is not as strong as that of ANNs. For example, a spike train composed of three spike values (each value can be 0 or 1) can only express 2^3 possible values. while the information encoding capacity of high-precision numerical data is almost unlimited (2^32). Fortunately, spike trains (SNN layer output) have both spatial and temporal dimensions, and expanding these dimensions can enhance the total information encoding capacity, i.e., “expressiveness”. The spatial dimension is constrained by the network structure, while the temporal dimension is controlled by the SNN time step T. Increasing the time step T can lead to an improvement in the information encoding capacity, thereby enhancing the “expressiveness” of SNN.
> “gradient accuracy” refers to the precision of the parameter gradients obtained by backpropagation. The reason for this notion arises from the fact that in the direct training of SNNs, the gradients computed by the backpropagation algorithm contain errors. These errors originate from the term $\frac{\partial s(i)}{\partial v(i)}$ in Eqn. 7, where this term is calculated as an approximated value by surrogate gradient during backpropagation because its original function is a step function. And there exists a certain error between the approximated value and the true value. As shown in Eqn. 7, as the time steps T increases, there are more instances of $\frac{\partial s(i)}{\partial v(i)}$, leading to a larger error in the calculated parameter gradient. [1] try to quantify the gradient error by the similarity between the gradient obtained through backpropagation (Eqn. 7) and the gradient computed by the finite differences (FDG).
>
> Due to the multidimensional nature of neural networks, even on the same training dataset, variations in training methods, and initial random seeds, the same networks may converge to various distinct local minima, leading to divergent performances on test data. The introduction of the concept of loss landscape provides an intuitive way to assess these local minima. Generally, sharper local minima indicate lower generalization of the network, while flatter local minima indicate higher generalization of the network.
> We think that “jump out of the local minimum point” and "moving surrounding it" do not contradict each other in the original sentence. They both refer to the idea of NMT making the SNN leave (Instead of falling into) a sharp local minimum. Eventually, the SNN reaches a flatter local minimum (See Fig. 9 in A.10), which ensures that even when changing the time step T, the SNN still cannot escape it.
>
>
> [1] Yuhang Li, Yufei Guo, Shanghang Zhang, Shikuang Deng, Yongqing Hai, and Shi Gu. Differentiable spike: Rethinking gradient-descent for training spiking neural networks. In A. Beygelzimer,Y. Dauphin, P. Liang, and J. Wortman Vaughan (eds.), Advances in Neural Information Processing Systems, 2021b. URL https://openreview.net/forum?id=H4e7mBnC9f0.

---

> > ### Comment · Reviewer_wYLe · 2023-11-22
> >
> > Thank you for your promising results on asynchronous hardware. It makes sense that flexibility to discrete time steps encourages robustness under asynchronous settings where timesteps do not exist, and it can be important empirical contribution to real applications of SNNs.
> >
> > As for theoretical parts, the “expressiveness” is quite different from the common expressive power of neural networks which measures the function approximation ability. And actually, if spiking time can be considered, SNNs even with a single spike can have strong expressive power. It is rate coding under discrete time steps that leads to the information capacity shortcoming you mentioned here. I would suggest changing another name for precision. Additionally, I think “gradient accuracy” is not well-defined here because “true value” for the derivative of step function is ill-defined. Finite difference is not “true gradient” and “gradient” is ill-defined under the discontinuous condition, if we consider loss based on the rate coding (gradients may exist if we consider temporal coding and BP through spiking time, but this paper considers rate-based loss). The authors may want to say that with more time steps, there is more discontinuity in the function under rate coding and it can be harder to train the model by gradient descent with “surrogate gradients” that try to approximate some terms similar to gradient.
> >
> > Overall, I suggest the authors to put more emphasis on the empirical advantages for real applications, e.g., discussion for asynchronization (currently I do not find it in the revised paper), and most of somewhat informal claims without rigorous justification may be moved to appendix for intuitive explanation, with better descriptions. This could enhance the presentation.

---

> > > ### Author Response · Authors · 2023-11-22
> > > **Response to Reviewer wYLe (Part 3)**
> > >
> > > Thank you for your professional, informative, and insightful replies and suggestions!
> > >
> > > We realize that "expressiveness" has already been defined and has an accurate meaning in previous studies, and in this context, the expressiveness of SNN on an asynchronous chip is irrelevant to the value of the simulation time step which deviates from our original meaning. Therefore our use of the term "expressiveness" was inaccurate, and therefore **we have replaced the word "expressiveness" in the original text and replaced it with "output precision" as you suggested**. Also, **We have removed the reference to "true value" in the previous revised paper** and **replaced the term "gradient error" with "gradient noise"**, since "true gradient" is not well-defined under the discontinuous condition.
> > >
> > > Finally, we **put the table of asynchronous tests into Section 5.2**. While we were pleased to find that TFSNN has good asynchrony-friendliness, limited time prevented us from designing it more specifically. Nevertheless, it is a promising direction, and we will continue to explore it in depth in our subsequent work to contribute to this area.

---

### Official Review · Reviewer_6ah9 · 2023-10-31

**Soundness:** 3 good
**Presentation:** 3 good
**Contribution:** 3 good
**Rating:** 6
**Confidence:** 4

**Summary:**

This work endeavors to adapt SNNs to different inference timesteps in a single training run. Mixed time step training (MTT) samples from a series of timestep setups for different stages of SNNs. Temporal transformation modules are inserted between stages to align input/output between neighboring stages. The results show it indeed improves the flexibility when facing variable-length input temporal patterns in both static-image and DVS datasets. The authors also provide an overall estimation of the accuracy when applying a group of different timestep setups.

**Strengths:**

The MTT performs a wider search along the timestep setting by sampling different. Compared to naive NMT, MTT urges each layer of SNNs to learn feature that is weakly correlated to input length from the previous layer. Such a strong regularization causes an issue in assessing mean and variance within BN layers but is immediately resolved by calibration. Overall, I believe MTT marks a significant step forward in the right direction.

**Weaknesses:**

However, all datasets used in MTT are directly retrieved or derived from static images (CIFAR10-DVS & CIFAR10, N-Caltech101 & Caltech101). I'm worried that these datasets contain essentially time-invariant features along the time dimension. Even for CIFAR10-DVS or N-Caltech101, only simple movement of static images is recorded using the event cameras. The authors should validate their methods on those datasets with affluent temporal dynamics, like from DVS-Gesture to audio such as GSC (Google SpeechCommands) or SHD (Spiking Heidelberg Dataset) to strengthen their results.

The former concern raises the other issue that, mixing different timesteps is partially empirically based on the finding that the temporal gradients resemble each other when timestep stretches or shrinks, and we expect the training will be stable to some extent since gradients are relatively similar. If the input has invariant length and a different amount of temporal information in essence, will the MTT work as it is now?

**Questions:**

Could the authors demonstrate results on datasets with rich temporal information? This could bring some real challenges to the TFSNN.

---

> ### Author Response · Authors · 2023-11-21
> **Response to Reviewer 6ah9**
>
> **Comment 1: The authors should validate their methods on those datasets with affluent temporal dynamics, like from DVS-Gesture to audio such as GSC (Google SpeechCommands) or SHD (Spiking Heidelberg Dataset) to strengthen their results.**
>
> Thanks for your valuable suggestion. We conducted experiments on DVS-Gesture, GSC, and SHD data respectively, and the research showed that TFSNN could function effectively as a time encoder, provided that the temporal configuration vectors used for training are monotonically non-increasing.
>
> For the DVS-Gesture, we use the sample network architecture(PLIF is replaced by standard LIF) and the same AER data pre-processing method as [1]. We train for 150 epochs and the result is shown in Table R1.
>
> **Table R1: DVS-Gesture**
> |PLIF|Our SDT|Our MTT|
> |-|-|-|
> |97.57|97.57|97.92|
>
>
> For the Spiking Heidelberg Digits, we adopt the plain 3-layer (l=3) feed-forward SNN architecture in [2], a fully connected SNN with an input width of 70 and 128 LIF neurons in each of the 3 hidden layers. The timestep of the first layer is fixed to the input timestep, while the timesteps of subsequent layers are restricted to monotonically non-increasing. We set $\tau = 0.9753$, which is equivalent to the parameter $\lambda$ in [2], namely $1 - 1 / \tau$ in most other articles, and train the model for 150 epochs. To ensure the validity of our results, we also reproduce the result using the code provided by [3]. The results are as follows. Notes that the l=2 results in [3] are read from Fig. 5 in [3].
>
> **Table R2: Spiking Heidelberg Digits**
> |results in [3] l=2|Repro with code in [3] l=3|Our SDT l=3|Our MTT l=3|
> |-|-|-|-|
> |60~70|75.26|74.43|79.68|
>
> For the Google Speech Command (GSC), we find its spiking version Spiking Speech Commands (SSC) [2], and test our method on SSC. SSC is a spiking dataset converted from Google Speech Commands v0.2 and is tailored for SNN. We continue using the same architecture and the same parameters as we used in SHD, except that here we only train the model for 60 epochs.
>
> **Table R3: Spiking Speech Commands**
> |Our SDT l=3|Our MTT l=3|
> |-|-|
> |57.75|60.15|
>
> We have also conducted experiments on seqMNIST in our original paper. seqMNIST is a reconstruction of MNIST by flattening the 2D images to 1D vectors spanning across time dimension. The 1D vectors are then fed into an SNN with only 1 input position and $28\times 28$ timesteps. The results are shown in Appendix A.12. For convenience, we paste this table below.
>
> **Table R4: seqMNIST (from A.12)**
> |Our RNN|SNN SDT|SNN MTT|
> |-|-|-|
> |56.22|55.75|64.56|
>
> As demonstrated above, experiments conducted on the four datasets thoroughly validate TFSNN's ability to handle datasets rich in temporal information.
>
> -------------
>
> **Comment 2: The former concern raises the other issue that, mixing different timesteps is partially empirically based on the finding that the temporal gradients resemble each other when the timestep stretches or shrinks, and we expect the training will be stable to some extent since gradients are relatively similar. If the input has invariant length and a different amount of temporal information in essence, will the MTT work as it is now?**
>
> Thank you for your insightful question. Indeed, our approach is based on the finding that the temporal gradients resemble each other when timestep stretches or shrinks, but is not dependent on it.
> For those datasets with affluent temporal dynamics, the input has a fixed length but with different temporal information in essence. As our experimental results demonstrate, MTT performs generally better than SDT. We believe this is because MTT allows a flexible framing strategy. In standard direct training, the temporal stream is divided by a fixed number of frames, and then the network is optimized for this specific number of frames. This can suffer from serious overfitting problems, e.g. training with T=10 and performing poorly with T=20. However, in MTT, loss only decreases when the network performs well across various frame numbers, which fosters robust generalization and results in superior overall performance.
>
> -------------
>
> [1] Wei Fang, Zhaofei Yu, Yanqi Chen, Timothee Masquelier, Tiejun Huang, and Yonghong Tian. Incorporating learnable membrane time constant to enhance learning of spiking neural networks. In Proceedings of the IEEE/CVF International Conference on Computer Vision (ICCV), pages 2661–2671, 2021.
>
> [2] Cramer, Benjamin, et al. "The heidelberg spiking data sets for the systematic evaluation of spiking neural networks." IEEE Transactions on Neural Networks and Learning Systems 33.7 (2020): 2744-2757.
>
> [3] Hammouamri, Ilyass, Ismail Khalfaoui-Hassani, and Timothée Masquelier. "Learning delays in spiking neural networks using dilated convolutions with learnable spacings." arXiv preprint arXiv:2306.17670 (2023).

---

> > ### Comment · Reviewer_6ah9 · 2023-11-23
> >
> > I would like to express my gratitude to the authors for their hard work in demonstrating the results on various datasets with rich temporal dependencies. It is evident that MTT outperforms SDT by a significant margin. As mentioned by the author in the text, NMT should be considered as a more valid baseline method of time-scale mixture training. It would be helpful to see the performance of NMT on these datasets as well.
> >
> > Furthermore, I suggest including the cosine similarity of the gradient under different time steps for the newly added datasets. This will provide clarity on the extent to which MTT relies on gradient similarity. If the gradients are significantly different, it would indicate that MTT is not dependent on gradient similarity, which would be a promising finding.

---

> ### Author Response · Authors · 2023-11-23
>
> As the discussion deadline is drawing near, could you please consider reviewing our response and reconsidering our work, if convenient for you? We're open to any additional questions and would be delighted to provide further clarification. It would be wonderful if we could address any concerns you may have. Thank you for your time and valuable feedback!

---

> ### Author Response · Authors · 2023-11-23
> **Response to Reviewer 6ah9 (Part 2)**
>
> Thank you for your professional and valuable advice. We conducted tests of NMT on the SHD dataset, but regrettably, due to time constraints, we were unable to test on other datasets and calculate the gradient similarity. Considering NMT as a specific type of MTT, we gradually converted MTT to NMT, and for convenience of description, we refer to the intermediate state of this conversion process as 'NMTT'.
>
> **Table R5 Spiking Heidelberg Digits**
> |Methods|Accuracy|
> |-|-|
> |MTT l=3 (t1=Tin>t2>t3) |79.68|
> |"NMTT" l=3 (t1=Tin>t2=t3) |79.42|
> |NMT l=3 (t1=t2=t3)|77.25|
>
> From Table R5, it is evident that NMT itself serves as a more valid baseline method compared to SDT. However, it proves to be weaker than the MTT method we proposed. (According to your suggestion, The results of NMT will be included in Table R1-R4 in the final version of the paper.) Here, t1/t2/t3 denotes the sampled timestep of the 1-st/2-nd/3-rd layer. For MTT and NMTT, we didn’t insert TTM between the input and the first layer since we directly make the first layer an encoding layer and let t1=Tin, where Tin is the timestep of the input. For NMT, we insert a TTM between the first layer and the input, so we can set t1=t2=t3.

---

### Author Response · Authors · 2023-11-21
**Global Response**

Dear reviewers, we sincerely appreciate your thorough review of the article and the valuable comments, suggestions, and questions you shared. We apologize for the delayed response, which was due to the execution of a series of necessary experiments.

**We have incorporated the necessary changes into the latest version of our paper, and we've highlighted them in blue for your convenience.**

Below, we outline some significant modifications prompted by your feedback.

------

 To facilitate a comprehensive overview, we have gathered all experimental results from the responses here.

**Table R1: Test Accuracy of NMNIST on GPU and asynchronous chip**
|Methods|GPU|Speck2e Devkit|
|-|-|:-:|
|SDT|98.61|96.18|
|MTT|97.79|98.5|

Notes to Table R1: Since the performance of TFSNN is largely decoupled from the influence of the time step due to the design of MTT and the concept of timestep also doesn't exist on asynchronous chips, TFSNN is naturally more suitable for deployment on asynchronous SNN chips. To demonstrate this capability, We deployed the MTT-trained SNN on Synsense's Speck2e Devkit for asynchronous testing. Specifically, we trained two networks on the NMNIST dataset using SDT and MTT, respectively, and then deployed them on the Speck2e Devkit, as detailed in Table R1.

**Table R2: DVS-Gesture**
|PLIF|Our SDT|Our MTT|
|-|-|-|
|97.57|97.57|97.92|

**Table R3: Spiking Heidelberg Digits**
|results in [4] l=2|Repro with code in [4] l=3|Our SDT l=3|Our MTT l=3|
|-|-|-|-|
|60~70|75.26|74.43|79.68|

**Table R4: Spiking Speech Commands**
|Our SDT l=3|Our MTT l=3|
|-|-|
|57.75|60.15|

-------

Thanks to the suggestions of reviewers 1Uof and wYLe, we find that we have not provided clear definitions for the words “expressiveness” and “gradient accuracy” in the manuscripts.

In the new version, we have included the following sentence to prove definitions for the two words (See the blue highlighted text in Sec 4.2.):

“expressiveness” refers to the information encoding capability of the spike trains. Due to the binary nature of the spike-based inter-layer information transmission in SNNs, their information encoding capacity is not as strong as that of ANNs. For example, a spike train composed of three spike values (each value can be 0 or 1) can only express 2^3 possible values. while the information encoding capacity of high-precision numerical data is almost unlimited (2^32). Fortunately, spike trains (SNN layer output) have both spatial and temporal dimensions, and expanding these dimensions can enhance the total information encoding capacity, i.e., “expressiveness”. The spatial dimension is constrained by the network structure, while the temporal dimension is controlled by the SNN time step T. Increasing the time step T can lead to an improvement in the information encoding capacity, thereby enhancing the “expressiveness” of SNN.

“gradient accuracy” refers to the precision of the parameter gradients obtained by backpropagation. The reason for this notion arises from the fact that in the direct training of SNNs, the gradients computed by the backpropagation algorithm contain errors. These errors originate from the term $\frac{\partial s(i)}{\partial v(i)}$ in Eqn. 7, where this term is calculated as an approximated value by surrogate gradient during backpropagation because its original function is a step function. And there exists a certain error between the approximated value and the true value. As shown in Eqn. 7, as the time steps T increases, there are more instances of $\frac{\partial s(i)}{\partial v(i)}$, leading to a larger error in the calculated parameter gradient. [1] try to quantify the gradient error by the similarity between the gradient obtained through backpropagation (Eqn. 7) and the gradient computed by the finite differences (FDG).

------

[1] Yuhang Li, Yufei Guo, Shanghang Zhang, Shikuang Deng, Yongqing Hai, and Shi Gu. Differentiable spike: Rethinking gradient-descent for training spiking neural networks. In A. Beygelzimer,Y. Dauphin, P. Liang, and J. Wortman Vaughan (eds.), Advances in Neural Information Processing Systems, 2021b. URL https://openreview.net/forum?id=H4e7mBnC9f0.

[2] Wei Fang, Zhaofei Yu, Yanqi Chen, Timothee Masquelier, Tiejun Huang, and Yonghong Tian. Incorporating learnable membrane time constant to enhance learning of spiking neural networks. In Proceedings of the IEEE/CVF International Conference on Computer Vision (ICCV), pages 2661–2671, 2021.

[3] Cramer, Benjamin, et al. "The heidelberg spiking data sets for the systematic evaluation of spiking neural networks." IEEE Transactions on Neural Networks and Learning Systems 33.7 (2020): 2744-2757.

[4] Hammouamri, Ilyass, Ismail Khalfaoui-Hassani, and Timothée Masquelier. "Learning delays in spiking neural networks using dilated convolutions with learnable spacings." arXiv preprint arXiv:2306.17670 (2023).

---

> ### Author Response · Authors · 2023-11-22
> **Global Response 2**
>
> Thanks to reviewer wYLe's valuable suggestions, according to which we made 2 extra edits to our paper.
>
> **1. We added a new paragraph to Sec 5.2 in our latest paper**
>
> **Compatibility with Asynchronous Chip** Since the performance of TFSNN is largely decoupled from the influence of the time step due to the design of MTT and the concept of timestep also doesn't exist on asynchronous chips, TFSNN is naturally more suitable for deployment on asynchronous SNN chips. To demonstrate this capability, we deployed the MTT-trained SNN on Synsense's Speck2e Devkit for asynchronous testing. Specifically, we trained two networks on the NMNIST dataset using SDT and MTT, respectively, and then deployed them on the Speck2e Devkit, as detailed in Table 7. As a result, the SDT-trained model suffered a performance drop of about 2\% while the MTT-trained model maintained and even achieved a higher performance on Speck2e Devkit.
>
> **Table 7: NMNIST Test Accuracy on Asynchronous Chip**
> |Methods|GPU|Speck2e Devkit|
> |-|-|-|
> |SDT|98.61|96.18(-2.43)|
> |MTT|97.79|98.5(+0.71)|
>
> -------
>
> **2. To avoid confusion, we replaced "expressiveness" with "output precision", deleted the reference to "true value" in the previous revised paper, and replaced the term "gradient error" with "gradient noise"** The changes can be seen in our latest revision.

---

### Meta-Review · Area_Chair_Kypm · 2023-12-11

**Metareview:**

The paper presents a novel approach to training Spiking Neural Networks (SNNs) by introducing a new method: Mixed Time Step Training (MTT). The authors claim this approach overcomes the limitations of current direct training methodologies, which are optimized for specific time steps, leading to computational inefficiencies in generalization. Experimental results, including accuracy rates on CIFAR10, CIFAR100, and ImageNet datasets, are promising.

However, the paper is reported as difficult to follow, with issues in organization, presentation and clarity. Concerns about the practicality of the proposed methods on neuromorphic hardware and their compatibility with different time steps for different blocks were raised. Theoretical analysis in the paper is limited, with several informal claims lacking rigorous justification.

**Justification For Why Not Higher Score:**

While the authors made efforts to address the raised concerns in their rebuttal, the revisions and clarifications provided are not sufficient to fully resolve the issues raised by the reviewers.

Further, the authors violated the anonymity policy by revealing their names in a private comment to the AC.

**Justification For Why Not Lower Score:**

N/A

---

### Decision · Program_Chairs · 2024-01-16

Reject